# Synthesis and Investigation of Flavanone Derivatives as Potential New Anti-Inflammatory Agents

**DOI:** 10.3390/molecules27061781

**Published:** 2022-03-09

**Authors:** Cynthia Sinyeue, Mariko Matsui, Michael Oelgemöller, Frédérique Bregier, Vincent Chaleix, Vincent Sol, Nicolas Lebouvier

**Affiliations:** 1Institut des Sciences Exactes et Appliquées (ISEA) EA7484, Campus de Nouville, Université de la Nouvelle Calédonie, Noumea 98851, New Caledonia; cynthia.sinyeue@etudiant.unc.nc (C.S.); mmatsui@pasteur.nc (M.M.); 2Laboratoire PEIREINE UR 22722, Faculté des Sciences et Techniques, 87060 Limoges, France; frederique.bregier@unilim.fr (F.B.); vincent.chaleix@unilim.fr (V.C.); vincent.sol@unilim.fr (V.S.); 3Group Immunity and Inflammation (GIMIN), Institut Pasteur of New Caledonia, Pasteur Network, Noumea 98845, New Caledonia; 4College of Science and Engineering, Discipline of Chemistry, James Cook University, Townsville, QLD 4811, Australia; michael.oelgemoeller@hs-fresenius.de; 5Faculty of Chemistry and Biology, Hochschule Fresenius gGmbH—University of Applied Science, 65510 Idstein, Germany

**Keywords:** flavanone, anti-inflammatory activity, structure–activity relationship (SAR), RAW264.7, pinocembrin

## Abstract

Flavonoids are polyphenols with broad known pharmacological properties. A series of 2,3-dihydroflavanone derivatives were thus synthesized and investigated for their anti-inflammatory activities. The target flavanones were prepared through cyclization of 2′-hydroxychalcone derivatives, the later obtained by Claisen–Schmidt condensation. Since nitric oxide (NO) represents an important inflammatory mediator, the effects of various flavanones on the NO production in the LPS-induced RAW 264.7 macrophage were assessed in vitro using the Griess test. The most active compounds were flavanone (**4G**), 2′-carboxy-5,7-dimethoxy-flavanone (**4F**), 4′-bromo-5,7-dimethoxy-flavanone (**4D**), and 2′-carboxyflavanone (**4J**), with IC50 values of 0.603, 0.906, 1.030, and 1.830 µg/mL, respectively. In comparison, pinocembrin achieved an IC_50_ value of 203.60 µg/mL. Thus, the derivatives synthesized in this work had a higher NO inhibition capacity compared to pinocembrin, demonstrating the importance of pharmacomodulation to improve the biological potential of natural molecules. SARs suggested that the use of a carboxyl-group in the *meta*-position of the B-ring increases biological activity, whereas compounds carrying halogen substituents in the *para*-position were less active. The addition of methoxy-groups in the *meta*-position of the A-ring somewhat decreased the activity. This study successfully identified new bioactive flavanones as promising candidates for the development of new anti-inflammatory agents.

## 1. Introduction

Inflammation is involved in many diseases, such as infectious diseases, chronic inflammation, asthma, diabetes, neurodegenerative diseases, coronaviruses, and cancer [1,2]. Anti-inflammatory treatments are mostly based on corticoids or non-steroidal anti-inflammatory drugs (NSAIDs), such as aspirin (2-acetyloxybenzoic acid) and ibuprofen (2-[4-(2-methylpropyl)phenyl]propanoic acid). However, long-term therapy involving these common pharmaceuticals leads to severe side effects such as gastrointestinal ulceration and bleeding, osteoporosis, hypertension, and glaucoma. Consequently, there is a need for new anti-inflammatory compounds with fewer or no side effect [3]. Macrophages are essential in the inflammatory process, and some bacterial endotoxins, such as lipopolysaccharide (LPS), allow their activation [4]. Nitric oxide (NO) is an inflammatory mediator that influences various biological processes [5]. At high levels, NO can exhibit cytotoxicity and tissue damage. Therefore, inhibition of this mediator can provide therapeutic effects and allow measurement of the degree of inflammation. Natural products such as the sesquiterpene yomogin isolated from *Artemisia princeps* or *Ginkgo biloba* extract are known to inhibit NO production [6,7].

Flavonoids are widely present in plants. A nutrition study revealed that diets rich in fruits and vegetables can help to prevent inflammatory diseases. These beneficial properties have been linked to synergetic effects of bioactive compounds, including flavonoids [8,9]. The latter compounds act as regulators of metabolic processes and have thus been investigated for the treatment of diseases. Consequently, flavonoids have been widely studied for their beneficial effects on human health, including antiallergenic, anti-inflammatory, vasodilating, anti-COVID-19, and antitumor capacities [1,2,10,11]. Flavonoids can furthermore act on important mechanisms in inflammatory processes by controlling regulatory enzymes and transcription factors [12]. Flavanones are a subgroup of flavonoids characterized by two aromatic rings (A and B) linked by a dihydropyrone ring C (Figure 1).

Pinocembrin (5,7-dihydroxyflavanone) is a natural flavanone present in fruits, spices, propolis, tea, and red wine and has shown beneficial properties, including anti-inflammatory potential [13,14]. A recent in vivo study proved the ability of pinocembrin to inhibit the LPS-stimulated inflammatory response in macrophages and to regulate the TLR4-NF-κB signaling pathway [15]. Furthermore, pinocembrin is also found in pine species that are exploited for their wood [16]. The easy availability of biomass and low cost of co-products from these species may thus provide a renewable access to pinocembrin and its derivatives.

In order to obtain pinocembrin analogues, a two-step synthesis strategy was followed in this study. The chalcones obtained through initial Claisen–Schmidt condensation were subjected to subsequent cyclization to their corresponding flavanones [17,18]. The flavanone derivatives obtained were subsequently examined as regulators for the NO production on the LPS-induced murine macrophage. The aim of this work was to investigate the structural requirements on the A- and B-rings of flavanones for anti-inflammatory activity. Simple structure–activity relationship (SAR) analysis consequently determined the most effective candidates for future optimization studies.

## 2. Results and Discussion

### 2.1. Synthesis

A series of 2′-hydroxy chalcones (**3A–3L**) were synthesized by Claisen–Schmidt condensation from appropriately substituted 2′-hydroxyacetophenones and benzaldehydes. Various experimental conditions, such as acid catalysis [19], base catalysis [20], heat treatment [21], and light activation [22,23], have been reported to achieve the subsequent cyclization to flavanones. Two cyclization procedures were chosen in this work: basic and photochemical activation (Figure 2). The photochemical route was investigated as a milder approach [24]; however, the efficiency of photocyclization is known to depend on the irradiation wavelength [25]. Commercial pinocembrin (PC) was selected as a reference molecule.

Following these synthetic approaches, a series of flavanones was prepared with or without methoxy-groups attached to the A-ring, while the B-ring carried various substituents in different positions (Table 1). The isolated yields of chalcones **3** ranged from 23–92%, while those of flavanones **4** varied widely between 7 and 74%, indicating a need for further optimization. After prolonged irradiation for 7 days, the photochemical cyclization gave a low isolated yield of just 7% for **4A**, despite a near-complete conversion of **3A** of 98%, suggesting significant losses during repeated purification by column chromatography. All other derivatives **4B–F** were obtained by base-catalyzed cyclization instead. For the flavanones containing two methoxy-groups on the A-ring (**A** to **F**), the highest yield of flavanones overall was obtained with the carboxyl-group (**4F**), followed by the methoxy substituent (**4B**). For the series without substitution on the A-ring (**G** to **K**), the highest flavanone yield was achieved with the carboxyl-group (**4J**), followed by the parent flavanone **4G**. Thus, when the corresponding benzaldehyde was substituted with an electron-withdrawing group at the 2-position (as in **4F** and **4J**), the formation of flavanones was effective. In contrast, the addition of electron-donating groups on the A-ring, as for derivatives **4****A**, **G,** and **L**, gave the desired flavanones in only low yields.

Both cyclization methods were found to have limitations. The photochemical method did not require the addition of base and proceeded cleanly, but it demanded an exhaustive irradiation time to achieve a high conversion. The basic activation was found to operate faster but led to by-products that required subsequent purification steps.

### 2.2. Evaluation of Biological Activities

#### 2.2.1. Biological Activities of Commercial Pinocembrin

Pinocembrin (PC) inhibitory activity was previously investigated on nitrite produced by LPS-induced RAW264.7 [26], and an IC_50_ of 203.60 µg/mL was determined. The anti-inflammatory bioactivity of commercially available PC was thus studied (Figure 3A), and the results showed a significant inhibitory response at 200 µg/mL (79.36 ± 7.30%) on LPS-dependent NO production, hence confirming the anti-inflammatory effect of PC in this concentration range. No inhibitory effect of PC was observed at 2 or 20 µg/mL (Figure 3A). The cytotoxicity was also analyzed (Figure 3B), and the results revealed a significant increase in percentage of cytotoxicity for PC at 200 µg/mL (94.23 ± 3.72%) compared to LPS-related cytotoxicity (32.28 ± 10.01%), while no significant difference in cytotoxicity was observed at 2 (25.76 ± 14.41%) and 20 µg/mL (34.45 ± 7.25%). A cytotoxic concentration of 200 µg/mL could possibly affect the determination of a reliable inhibitory response on NO production, as lower quantities of viable cells could result in inherently lower NO levels. These results thus highlight the importance of evaluating the cytotoxicity of natural compounds and their derivatives.

#### 2.2.2. Cytotoxicity of Flavanone Derivatives

Due to the cytotoxicity of PC, the cytotoxicity to LPS-induced RAW264.7 at 2 µg/mL was evaluated for all flavanone derivatives (Figure 4). Initial cytotoxicity of untreated cells was calculated at 32.41 ± 3.4%, which is consistent with previous observations [27]. No difference with LPS-related cytotoxicity was determined at 33.26 ± 10.61%, and no significant increase was observed for any flavanone at 2 µg/mL, with percentages of cytotoxicity ranging between 20 and 40%. Thus, the potential inhibitory effect of these compounds was investigated at this concentration and compared to PC.

#### 2.2.3. Inhibitory Activity on LPS-Induced NO Production

The anti-inflammatory responses of all synthetized flavanone compounds at 2 µg/mL were examined by evaluating their inhibitory effect on NO induction produced by murine macrophage RAW264.7 treated with bacterial LPS (Figure 5 and Table 2). The inhibitory response of the reference anti-inflammatory molecule dexamethasone at 100 nM was confirmed (100.4 ± 16.14%). As noted above, PC did not show an effective response at 2 µg/mL. In contrast, various inhibitory activities were determined for flavanones **4A–L** at the same concentration.

The most active compounds were **4F** (78.65 ± 24.73%), **4G** (75.65 ± 46.88%), **4J** (72.56 ± 22.70%) and **4L** (64.97 ± 42.37%), respectively, followed by **4D** (61.10 ± 25.50%), and **4****A** (58.99 ± 45.19%). Due to their large SD variation, the following compounds were considered less effective but still showed a significant inhibitory response: **4K** (73.29 ± 86.10%), **4B** (40.45 ± 49.71%), and **4E** (26.85 ± 24.43%). **4C** (19.14 ± 33.83%), **4H** (3.12 ± 34.64%), and **4I** (−0.87 ± 37.54%) were not significantly bioactive (Table 2).

The IC_50_ values for the most effective molecules **4F**, **4G**, **4J**, and **4D** were subsequently calculated (Table 3). Dexamethasone presented an IC_50_ of 0.005 µg/mL (95% CI: 0.003–0.008). The most active flavanones were **4G,** with the lowest IC_50_ of 0.603 µg/mL (95% CI: 0.366–1.003) and **4F** with an IC_50_ of 0.906 µg/mL (95% CI: 0.550–1.765), while **4D** and **4J** furnished lower inhibitory activities with higher IC_50_ values of 1.030 µg/mL (95% CI: 0.675–1.382) and 1.830 µg/mL (95% CI: 1.467–2.677), respectively.

### 2.3. Structure–Activity Relationship Study

The biological evaluation clearly indicated that the percentage of inhibition depended on the substitution pattern of flavanones **4A–L**; Figure 6 summarizes the pharmacomodulation obtained from this study.

For the first series with two methoxy-groups at the A-ring, the most active molecules were **4F** (2′-carboxy-5,7-dimethoxy-flavanone) and **4D** (4′-bromo-5,7-dimethoxy-flavanone). Hence, the presence of a carboxyl-group in the *ortho*-position or a bromo substituent in the *para*-position of the B-ring increased the anti-inflammatory activity. In addition, **4D** (4′-bromo-5,7-dimethoxy-flavanone) was more active than the related **4E** (4′-chloro-5,7-dimethoxy-flavanone). For the second series without substitution on the A-ring, the most active compounds were **4G** (flavanone), **4J** (2′-carboxyflavanone), and **4K** (5′-bromo-2′-methoxy-flavanone). Overall, the presence of methoxy-groups on the flavanone skeleton impacted the NO inhibitory activity. Indeed, flavanones with a methoxy-group in the 5-position demonstrated strong inhibitory activity on NO production from LPS-stimulated macrophage cells [28]. For the functionalization on the A-ring, the comparison of **4G** (flavanone), **4L** (5-methoxyflavanone), and **4A** (5,7-dimethoxyflavanone) showed that the addition of a methoxy-group contributes to a drop in activity. Additionally, the flavanone derivative with the methoxy-group in the 3-position (**4B**) was found more active than the corresponding analogue with the methoxy-substituent in the 4-position (**4C**). Pinocembrin (5,7-dihyodroxyflavanone) containing two hydroxyl-groups on the A-ring showed no significant effect at the same concentration if compared to the flavanone derivatives synthesized. These results are in agreement with Kim’s work, which demonstrated that naringenin (5,7,4′-trihydroxyflavanone) was inactive up to 100 µM [29]. It may thus be suggested that the presence of a hydroxy-group on the A-ring only weakly affects the NO inhibition.

This SARs clearly revealed that a 2′-carboxy-group is beneficial to achieve effective inhibition of NO production. This finding may be due to the electron-withdrawing and highly polar nature of the carboxyl-group. According to Shin’s study, a bulky and/or hydrophobic substituent in the *meta*-position of the B-ring results in the most active structure to inhibit NF-κB activation [30]. This study identified **4B** (3’-methoxyflavanone) as more active than **4C** (4’-methoxyflavanone) and revealed the importance of a carboxyl-group in the *ortho*-position of the B-ring instead, as in **4F** and **4J**.

The predictive analysis of the drug-like absorption of the most active compounds (Table 4) revealed that all compounds meet Lipinski’s rule of five [31]: hydrogen bond donors ≤ 5, hydrogen bond acceptors ≤ 10, molecular mass ≤ 500 daltons, an octanol-water partition coefficient (log *p*) ≤ 5, and a polar surface area ≤ 140 Å^2^. Subsequent future pharmacomodulations, however, may lead to compounds with further improved activity.

## 3. Materials and Methods

### 3.1. Syntheses

#### 3.1.1. General Information

All chemicals were purchased from Sigma–Aldrich (Saint-Louis, MO, USA) and were used as received. HPLC grade solvents were bought from Fischer Chemicals (Leicestershire, UK) and were used without further purification. Reactions were monitored by thin-layer chromatography (TLC) using silica gel-precoated aluminum sheets (60 F254, Merck KGaA, Darmstadt, Germany) with different solvent systems (cyclohexane/EtOAc or DCM/MeOH). Products were visualized with UV irradiation at 365 and 254 nm or by treatment with sulfuric acid-vanillin. Selected compounds were purified by column chromatography (CC) using silica gel 60 (0.015–0.040 mm, Merck) as a stationary phase or by LH-20 Sephadex column chromatography. Electrothermal and Gallenkamp melting point apparatuses (MP, OC, uncorrected) were used to determine the melting points in open capillaries. Nuclear magnetic resonance (NMR) spectra were recorded on a Bruker x400 (^1^H NMR: 400 MHz, ^13^C NMR: 100 MHz) spectrometer or Bruker DPX 500 NMR spectrometer (^1^H NMR: 500 MHz, ^13^C NMR: 125 MHz). The solvents used were CDCl_3_ or dimethyl sulfoxide (DMSO)-d_6_. Chemical shifts (δ) were recorded in parts per million (ppm) relative to residual solvent peaks (CDCl_3_: ^1^H NMR: δ = 7.26 ppm and ^13^C NMR: δ = 77.00 ppm; DMSO-d_6_: ^1^H NMR: δ = 2.50 ppm and ^13^C NMR: δ = 39.52 ppm). Coupling constants were reported in hertz (Hz). Multiplicities were reported as s (singlet), br s (broad singlet), d (doublet), t (triplet), br (broad), m (multiplet), dd (doublet of doublets), dt (doublet of triplets), ddt (doublet of doublet of triplet), ddd (doublet of doublets of doublets), app d (apparent doublet), and app t (apparent triplet). High resolution electrospray ionization mass spectrometry (HR ESI-MS) was performed at the ICOA/CBM platform (Orléans University) on a Bruker Q-TOF maXis mass spectrometer, coupled to an Ultimate 3000 RSLC chain (Dionex).

#### 3.1.2. Synthetic Methods

For characterization of the molecular structures, the numbering of atoms shown in Figure 7 was followed.

General Procedure for the synthesis of chalcones (**3A–L**).

A solution of MeOH (25 mL), the corresponding acetophenones **1A–C** (1 eq), the appropriate benzaldehyde **2A–G** (1 eq), and excess of NaOH were stirred at room temperature, and the progress of the reaction was monitored by TLC. Upon completion, the excess of NaOH was neutralized by addition of HCl (1 M) with pH control. The solvent was evaporated, the residue was taken up in EtOAc, and the resulting solution was washed with distilled water. The organic layer was dried with MgSO_4_ and filtered, and the solvent was evaporated under reduced pressure. The crude product was purified by crystallization from methanol (MeOH) or by column chromatography.

(E)-1-(2′-hydroxy-4,6-dimethoxyphenyl)-3-phenylprop-2-en-1-one or 2′-Hydroxy-4′,6′-dimethoxy-chalcone or flavokavain B (**3A**)

Prepared following the general procedure starting from 2-hydroxy-4,6-dimethoxy-acetophenone (**1****A**, 0.1458 g, 0.74 mmol) and benzaldehyde (**2A**, 0.078 g, 0.74 mmol) with 10 eq of NaOH. Yellow crystals (0.057 g, 0.20 mmol), 27% yield, m.p. (Gallenkamp apparatus): 89 °C (85–86 °C, [32]).

^1^H-NMR (400 MHz, CDCl_3_): δ 14.20 (s, 1H, 2′-OH), 7.83 (d, *J* = 15.7 Hz, 1H, β-H), 7.72 (d, *J* = 15.7 Hz, 1H, α-H), 7.54 (dd, *J* = 7.5 Hz, 2.1 Hz, 2H, 2-H, 6-H), 7.30 (m, 3H, 3-H, 4-H, 5-H), 6.05 (d, *J* = 2.4 Hz, 1H, 3′-H), 5.90 (d, *J* = 2.4 Hz, 1H, 5′-H), 3.86 (s, 3H, 4′-OCH_3_), 3.77 (s, 3H, 6′-OCH_3_). ^13^C-NMR (100 MHz, CDCl_3_): δ 166.5 (4′-C), 162.6 (2′-C), 142.3 (β-C), 130 (1-C), 128.89 (3-C, 6-C), 128.4 (5-C), 128.3 (2-C), 127.6 (4-C), 102.7 (1′-C), 91.4 (3′-C), 91.1 (5′-C), 55.7 (4′-OCH_3_), 55.5 (6′-OCH_3_). ^1^H and ^13^C spectral data were consistent with the literature [32].

(E)-1-(2′-hydroxy-4′,6′-dimethoxyphenyl)-3-(3-methoxyphenyl)prop-2-en-1-one or 2′-hydroxy-3, 4′,6′-trimethoxy-chalcone (**3B**)

Prepared following the general procedure starting from 2-hydroxy-4,6-dimethoxy-acetophenone (**1A**, 0.50 g, 2.55 mmol) and 3-methoxy-benzaldehyde (**2B**, 0.346 g, 2.55 mmol) with 10 eq of NaOH. Yellow orange crystals (0.655 g, 2.09 mmol), 82% yield, m.p. (Electrothermal apparatus): 100 °C (100 °C, [33]).

^1^H-NMR (500 MHz, CDCl_3_): δ 14.25 (s, 1H, 2′-OH), 7.87 (d, *J* = 15.7 Hz, 1H, β-H), 7.73 (d, *J* = 15.7 Hz, 1H, α-H), 7.32 (t, *J* = 7.9 Hz, 1H, 5-H), 7.21 (d, *J* = 7.6 Hz, 1H, 6-H), 7.12 (d, *J* = 1.2 Hz, 1H, 2-H), 6.93 (dd, *J* = 8,2 Hz, 1H, 4-H), 6.11 (d, *J* = 2.4 Hz, 1H, 3′-H), 5.96 (d, *J* = 2.4 Hz, 1H, 5′-H), 3.91 (s, 3H, 4′-OCH_3_), 3.85 (s, 3H, 6′-OCH_3_), 3.83 (s, 3H, 3-OCH_3_). ^13^C-NMR (125 MHz, CDCl_3_): δ 192.6 (CO), 168.4 (4′-C), 166.3 (6′-C), 162.5 (2′-C), 159.9 (3-C), 142.2 (β-C), 137.0 (1-C),129.8 (5-C), 127.9 (6-C), 120.9 (α-C), 115.6 (4-C), 113.7 (2-C), 106.4 (1′-C), 93.8 (3′-C), 91.3 (5′-C), 55.8 (3-OCH_3_), 55.6 (5-OCH_3_), 55.3 (7-OCH_3_). ^1^H and ^13^C spectral data were consistent with the literature [33,34].

(E)-1-(2′-hydroxy-4′.6′-dimethoxyphenyl)-3-(4-methoxyphenyl)prop-2-en-1-one or 2′-hydroxy-4,4′,6′-trimethoxy-chalcone (**3C**) has been previously synthesized by our research group [33].

(E)-3-(4-bromophenyl)-1-(2′-hydroxy-4′,6′-dimethoxyphenyl)prop-2-en-1-one or 2′-hydroxy-4-bromo-4′,6′-dimethoxy-chalcone (**3D**)

Prepared following the general procedure starting from 2-hydroxy-4,6-dimethoxy-acetophenone (**1A**, 0.216 g, 1.10 mmol) and 4-bromo-benzaldehyde (**2D**, 0.204 g, 1.10 mmol) with 2.5 eq of NaOH. Yellow crystals (0.206 g, 0.57 mmol), 52% yield, m.p. (Gallenkamp apparatus): 166 °C (165 °C, [33], 150–151 °C, [32]).

^1^H NMR (400 MHz, CDCl_3_): δ 14.23 (s, 1H, 2′-OH), 7.90 (d, *J* = 15.6 Hz, 1H, β-H), 7.72 (d, *J* = 15.6 Hz, 1H, α-H), 7.58–7.53 (m, 2H, 3-H, 5-H), 7.51–7.46 (m, 2H, 2-H, 6-H), 6.14 (d, *J* = 2.4 Hz, 1H, 3′-H), 5.99 (d, *J* = 2.4 Hz, 1H, 5′-H), 3.94 (s, 3H, 4′-OCH_3_), 3.87 (s, 3H, 6′-OCH_3_). ^13^C NMR (100 MHz, CDCl_3_): δ 192.5 (CO), 166.3 (4′-C), 162.9 (6′-C), 162.6 (2′-C), 140.9 (β-C), 132.13 (5-C, 3-C) 131.6 (1-C), 129.69 (2-C, 6-C), 128.2 (4-C), 124.2 (α-C), 106.3 (1′-C), 93.8 (3′-C), 91.4 (5′-C), 55.9 (4′-OCH_3_), 55.6 (6′-OCH_3_). ^1^H and ^13^C spectra were consistent with the literature [32].

(E)-3-(4-chlorophenyl)-1-(2′-hydroxy-4′,6′-dimethoxyphenyl)prop-2-en-1-one or 4-chloro-2′-hydroxy-4′,6′-dimethoxy-chalcone (**3E**)

Prepared following the general procedure starting from 2-hydroxy-4,6-dimethoxy-acetophenone (**1A**, 0.50 g, 2.55 mmol) and 4-chloro-benzaldehyde (**2E**, 0.358 g, 2.55 mmol) with 5 eq of NaOH. Yellow crystals (0.189 g, 0.60 mmol), 23% yield, m.p. (Gallenkamp apparatus): 164 °C (173–175 °C, [32], 2006 and 158–168 °C, [33]).

^1^H NMR (400 MHz, CDCl_3_): δ 14.24 (s, 1H, 2′-OH), 7.88 (d, *J* = 15.6 Hz, 1H, β-H), 7.74 (d, *J* = 15.6 Hz, 1H, α-H), 7.55 (d, *J* = 8.6 Hz, 2H, 2-H, 6-H), 7.44–7.36 (m, 2H, 3-H, 5-H), 6.14 (d, *J* = 2.4 Hz, 1H, 3′-H), 5.99 (d, *J* = 2.4 Hz, 1H, 5′-H), 3.94 (s, 3H, 4′-OCH_3_), 3.87 (s, 3H, 6′-OCH_3_). ^1^H spectral data were consistent with the literature [32].

(E)-2-(3-(2′-hydroxy-4′,6′-dimethoxyphenyl)-3-oxoprop-1-en-1-yl)benzoic acid or 2′-hydroxy-2-carboxy 4′,6′-dimethoxychalcone (**3F**)

Prepared following the general procedure starting from 2-hydroxy-4,6-dimethoxy-acetophenone (**1A**, 0.369 g, 1.89 mmol) and 2-carboxy-benzaldehyde (**2F**, 0.283 g, 1.89 mmol) with 10 eq of NaOH. Yellow solid (0.220 g, 0.67 mmol), 36% yield, m.p. (Gallenkamp apparatus): 161 °C (160–161 °C, [32]).

^1^H NMR (400 MHz, CDCl_3_): δ 14.04 (s, 1H, 2′-OH), 13.74 (s, 1H, 2-COOH),8.57 (d, *J* = 15.4 Hz, 1H, β-H), 8.10 (dd, *J* = 7.1 Hz, 1.0 Hz, 1H, 3-H), 7.91 (d, *J* = 15.4 Hz, 1H, α-H), 7.78 (m, 1H, 5-H), 7.66 (m, 1H, 6-H), 7.56 (td, *J* = 8.1 Hz, 1.5 Hz, 1H, 4-H), 6.15 (d, *J* = 2.4 Hz, 1H, 3′-H), 6.12 (d, *J* = 2.4 Hz, 1H, 5′-H), 4.00 (s, 3H, 4′-OCH_3_), 3.90 (s, 3H, 6′-OCH_3_).^1^H spectral data were consistent with the literature [32]. HRMS (ESI): Calc. for C_18_H_16_O_6_, [M + H]^+^
*m*/*z*: 329.10196, found: 329.10204, [M + Na]^+^ *m*/*z*: 345.07334, found: 345.07320.

(E)-1-(2′-hydroxyphenyl)-3-phenylprop-2-en-1-one or 2′-hydroxy-chalcone (**3G**)

Prepared following the general procedure starting from 2-hydroxy-acetophenone (**1B**, 1 g, 7.34 mmol) and benzaldehyde (**2A**, 0.78 g, 7.34 mmol) with 10 eq of NaOH. Yellow solid (1.013 g, 4.52 mmol), 62% yield, m.p. (Gallenkamp apparatus): 77 °C (44–150 °C CAS: 1214-47-7 Aldrich).

^1^H NMR (400 MHz, CDCl_3_): δ 12.82 (s, 1H, 2′-OH), 7.96 (d, *J* = 15.6 Hz, 1H, β-H), 7.97–7.94 (dd, *J* = 8.1 Hz, 1.6 Hz, 1H, 6′-H), 7.72–7.67 (m, 3H, 2-H, 3-H, 5-H), 7.69 (d, *J* = 15.6 Hz, 1H, α-H), 7.53 (td, *J* = 8.6 Hz, 1.6 Hz, 1H, 5′-H), 7.49–7.45 (m, 4H, 2′-H, 3′-H, 4′-H, 4-H), 7.06 (dd, *J* = 8.5 Hz, 0.9 Hz, 1H, 6-H), 6.98 (ddd, *J* = 8.3 Hz, 7.4 Hz, 1.0 Hz, 1H, 4′-H). ^13^C NMR (100 MHz100 MHz, CDCl_3_): δ 193.7 (CO), 163.6 (2′-C), 145.5 (β-C), 1 36.1 (1-C), 130.9 (4′-C), 129.6 (5′-C), 129.08 (3-C), 129.07 (5-C), 128.68 (2-C, 128.67 (6-C), 127.7 (4-C), 120.2 (α-C), 118.9 (3′-C), 118.7 (1′-C). HRMS (ESI): Calc. for C_15_H_12_O_2_, [M + H]^+^
*m*/*z*: 225.09101, found: 225.09112, [M + Na]^+^ *m*/*z*: 247.07295, found: 247.07318.

(E)-3-(4-chlorophenyl)-1-(2′-hydroxyphenyl) prop-2-en-1-one or 4-chloro-2′-hydroxy-chalcone (**3H**)

Prepared following the general procedure starting from 2-hydroxy-acetophenone (**1B**, 2 g, 14.69 mmol) and 4-chloro-benzaldehyde (**2E**, 2.06 g, 14.69 mmol) with 5 eq of NaOH. Yellow solid (3.265 g, 12.62 mmol), 86% yield, m.p. (Gallenkamp apparatus): 135–140 °C (149–150 °C, [35]).

^1^H NMR (400 MHz, CDCl_3_): δ 12.76 (s, 1H, 2′-OH), 7.93 (dd, *J* = 8.1 Hz, 1.6 Hz, 1H, 6′-H), 7.89 (d, *J* = 15.6 Hz, 1H, β-H), 7.65 (d, *J* = 15.6 Hz, 1H, α-H), 7.65–7.60 (m, 2H, 3-H, 5-H), 7.54 (ddd, *J* = 8.8 Hz, 7.3 Hz, 1.6 Hz, 1H, 5′-H), 7.47–7.41 (m, 2H, 3′-H, 6′-H), 7.06 (dd, *J* = 8.4 Hz, 0.9 Hz, 1H, 6-H), 6.98 (ddd, *J* = 8.5 Hz, 7.2 Hz, 1.1 Hz, 1H, 4′-H). ^13^C NMR (100 MHz, CDCl_3_): δ 193.5 (CO), 163.7 (2′-C), 143.9 (β-C), 136.6 (4-C), 133.1 (1-C), 129.79 (3-C), 129.78 (5-C), 129.6 (4′-C), 129.38 (2-C), 129.37 (6-C), 120.7 (1′-C), 119.9 (α-C), 118.9 (3′-C), 118.7 (5′-C) [34]. HRMS (ESI): Calc. for C_15_H_12_ClO_2_, [M + H]^+^ *m*/*z*: 259.05203, found: 259.05249.

(E)-1-(2′-hydroxyphenyl)-3-(4-methoxyphenyl)prop-2-en-1-one or 2′-hydroxy-4-methoxy-chalcone (**3I**)

Prepared following the general procedure starting from 2-hydroxy-acetophenone (**1B**, 2 g, 14.69 mmol) and 4-methoxy-benzaldehyde (**2C**, 2 g, 14.69 mmol) with 5 eq of NaOH. Orange solid (2.088 g, 8.21 mmol), 56% yield, m.p. (Gallenkamp apparatus): 84 °C (84–86 °C, [35]).

^1^H NMR (400 MHz, CDCl_3_): δ 12.95 (s, 1H, 2′-OH), 7.95 (dd, *J* = 8.1 Hz, 1.6 Hz, 1H, 6′-H), 7.93 (d, *J* = 15.4 Hz, 1H, β-H), 7.68–7.63 (m, 2H, 3-H, 5-H), 7.57 (d, *J* = 15.4 Hz, 1H, α-H), 7.51 (ddd, *J* = 8.6 Hz, 7.3 Hz, 1.6 Hz, 1H, 5′-H), 7.05 (dd, *J* = 8.4 Hz, 0.9 Hz, 1H, 6-H), 7.00–6.96 (m, 2H, 2-H, 3′-H), 6.99–6.93 (m, 1H, 4′-H), 3.89 (s, 3H, 4-OCH_3_). ^13^C NMR (100 MHz, CDCl_3_): δ 193.7 (CO), 163.6 (2′-C), 162.1 (4-C), 145.4 (β-C), 136.1 (6′-C), 130.55 (1-C), 130.54 (3-C), 130.53 (5-C), 129.55 (4′-C), 129.54 (2-C), 127.4 (6-C), 120.2 (1′-C), 118.8 (α-C), 118.6 (5′-C), 117.6 (3′-C), 55.5 (4-OCH_3_). HRMS (ESI): Calc. for C_16_H_14_O_3_, [M + H]^+^ *m*/*z*: 255.10157, found: 255.10208, [M + Na]^+^ *m*/*z*: 277.08352 found: 277.08381.

(E)-2-(3-(2′-hydroxyphenyl)-3-oxoprop-1-en-1-yl)benzoic acid or 2′-hydroxy-2-carboxy-chalcone (**3J**)

Prepared following the general procedure starting from 2-hydroxy-acetophenone (**1B**, 2.00 g, 14.69 mmol) and 2-carboxy-benzaldehyde (**2F**, 2.20 g, 14.69 mmol) with 5 eq of NaOH. Yellow solid (1.276 g, 4.76 mmol), 32% yield, m.p. (Gallenkamp apparatus): 144 °C.

^1^H NMR (400 MHz, CDCl_3_): δ 12.03 (s, 1H, 2′-OH), 7.96 (dd, *J* = 7.8 Hz, 2.3 Hz, 1H, 6′-H), 7.71 (ddd, *J* = 8.4 Hz, 7.4 Hz, 0.9 Hz, 1H, 5-H), 7.67 (dd, *J* = 8.1 Hz, 1.5 Hz, 1H, 6-H), 7.62 – 7.57 (m, 2H, 3-H, 4-H), 7.54 (ddd, *J* = 8.6 Hz, 7.3 Hz, 1.5 Hz, 1H, 5′-H), 7.05 (dd, *J* = 8.5 Hz, 0.7 Hz, 1H, 3′-H), 6.93 (ddd, *J* = 8.1 Hz, 7.1 Hz, 0.9 Hz, 1H, 4′-H), 6.19 (t, *J* = 6.4 Hz, 1H), 3.81 (dd, *J* = 17.6 Hz, 6.2 Hz, 1H, β-H), 3.46 (dd, *J* = 17.6 Hz, 6.8 Hz, 1H, α-H). ^13^C NMR (100 MHz, CDCl_3_): δ 201.7 (CO), 169.9 (2-C, COOH), 162.6 (2′-C), 149.3 (β-C), 137.2 (1-C), 129.84 (3-C), 129.83 (4′-C), 129.82 (5′-C), 129.63 (4-C), 129.62 (5-C), 125.93 (6′-C), 125.92 (6-C), 122.6 (2-C), 119.3 (1′-C), 119.1 (α-C), 118.8 (3′-C). HRMS (ESI): Calc. for C_16_H_12_O_4_, [M + H]^+^ *m*/*z*: 269.08084, found: 269.08139, [M + Na]^+^ *m*/*z*: 291.06278, found: 291.06338.

(E)-3-(5-bromo-2-methoxyphenyl)-1-(2′-hydroxyphenyl) prop-2-en-1-one or 5-bromo-2′-hydroxy-2-methoxy-chalcone (**3K**)

Prepared following the general procedure starting from 2-hydroxy-acetophenone (**1B**, 1.00 g, 7.34 mmol) and 5-bromo-2-methoxy-benzaldehyde (**2G**, 1.58 g, 7.34 mmol) with 10 eq of NaOH. Yellow solid (2.056 g, 6.17 mmol), 84% yield, m.p. (Gallenkamp apparatus): 121 °C.

^1^H NMR (400 MHz, CDCl_3_): δ 12.84 (s, 1H, 2′-OH), 8.16 (d, *J* = 15.7 Hz, 1H, β-H), 7.95 (dd, *J* = 8.1 Hz, 1.5 Hz, 1H, 6′-H), 7.78 (d, *J* = 2.4 Hz, 1H, 6-H), 7.74 (d, *J* = 15.7 Hz, 1H, α-H), 7.56–7.48 (m, 2H, 5′-H, 3′-H), 7.06 (dd, *J* = 8.4 Hz, 0.9 Hz, 1H, 4-H), 6.98 (td, *J* = 8.2 Hz, 7.2 Hz, 1.2 Hz, 1H, 4′-H), 6.87 (d, *J* = 8.9 Hz, 1H, 3-H), 3.94 (s, 3H, 2-OCH_3_). ^13^C NMR (100 MHz, CDCl_3_): δ 193.9 (CO), 163.6 (2′-C), 157.9 (2-C), 139.2 (β-C), 136.4 (1-C), 134.4 (4-C), 131.4 (6-C), 129.75 (4′-C), 129.74 (5′-C), 125.7 (6′-C), 121.7 (α-C), 120.1 (1′-C), 118.9 (5-C), 118.6 (3′-C), 113.1 (3-C), 55.9 (2-OCH3). HRMS (ESI): Calc. for C_16_H_13_BrO_3_, [M + H]^+^ *m*/*z*: 333.01208, found: 333.01247, [M + Na]^+^ *m*/*z*: 354.99403, found: 354.99467.

(E)-1-(2′-hydroxy-6-methoxyphenyl)-3-phenylprop-2-en-1-one or 2′-hydroxy-6′-methoxy-chalcone (**3L**)

Prepared following the general procedure starting from 2-hydroxy-6-methoxy-acetophenone (**1C****,** 0.5 g, 3.0 mmol) and benzaldehyde (**2A**, 0.32 g, 3.0 mmol) with 10 eq of NaOH. Yellow crystals (0.657 g, 2.58 mmol), 30% yield, m.p. (Electrothermal apparatus): 100 °C.

^1^H NMR (500 MHz, CDCl_3_): δ 13.13 (s, 1H, 2′-OH), 7.86 (d, *J* = 15.6 Hz, 1H, β-H), 7.80 (d, *J* = 15.6 Hz, 1H, α-H), 7.62–7.58 (m, 2H, 3-H, 5-H), 7.43–7.36 (m, 3H, 2-H, 4-H, 6-H), 7.34 (t, *J* = 8.1 Hz, 1H, 4′-H), 6.60 (d, *J* = 8.2 Hz, 1H, 5′-H), 6.41 (d, *J* = 8.3 Hz, 1H, 3′-H), 3.93 (s, 3H, 6′-OCH_3_). ^13^C NMR (125 MHz, CDCl_3_): δ 194.5 (4-C, CO), 164.9 (2′-C), 161.0 (6′-C), 142.9 (β-C),136.0 (4′-C), 135.3 (1-C), 130.3 (4′-C), 128.45 (3-C, 5-C), 127.6 (1′-C), 126.8 (2-C, 6-C), 112.1 (3′-C), 112.0 (5′-C),101.6 (α-C), 79.4 (β-H), 55.9 (6′-OCH_3_).

Flavanone **3A** synthesis by photochemical activation.

Chalcone **3A** (85 mg, 1 eq) was dissolved in 70 mL of ethanol in a Pyrex vessel. The solution was purged with purified nitrogen gas for 10 min before the reaction vessel was sealed. Irradiation was conducted in an RPR-200 photochemical reactor equipped with 16 fluorescent tubes (8 W each, 419 ± 25 nm) at approximately 30 °C [36]. The progress of the reaction was followed by TLC analysis, and after 7 days, almost complete conversion to the desired cyclized product was observed. The solution was evaporated to dryness, and the residue was purified on a chromatographic column.

5,7-dimethoxy-2-phenyl-2,3-dihydro-4H-benzopyran-4-one or 5,7-dimethoxy-2-phenylchroman-4-one or 5,7-dimethoxy-flavanone (**4A**)

This product was purified by repeated column chromatography (C_6_H_12_/EtOAc 90:10, 70:30 and 50:50). White solid (0.006 g, 0.02 mmol), 7% yield, m.p. (Gallenkamp apparatus): 144–146 °C.

^1^H NMR (400 MHz, CDCl_3_): δ 7.41–7.26 (m, 5H, 2′-H, 3′-H, 4′-H, 5′-H, 6′-H), 6.10 (d, *J* = 2.3 Hz, 1H, 6-H), 6.03 (d, *J* = 2.3 Hz, 1H, 8-H), 5.35 (dd, *J* = 13.1 Hz, 3.0 Hz, 1H, 2-H), 3.83 (s, 3H,5-OCH3), 3.76 (s, 3H, 7-OCH3), 2.96 (dd, *J* = 16.6 Hz, 13.1 Hz, 1H, 3ax-H), 2.74 (dd, *J* = 16.6 Hz, 3.1 Hz, 1H, 3eq-H). ^13^C NMR (100 MHz, CDCl_3_): δ 192.7 (4-C, CO), 166.26 (7-C), 165.9 (5-C), 162.5 (8a-C), 135.6 (1′-C), 128.45 (3′-C, 5′-C), 127.7 (4′-C), 126.85 (2′-C, 6′-C), 106.4 (4a-C), 93.8 (8-C), 93.7 (6-C), 79.5 (2-C), 56.8 (7-OCH3), 55.9 (4-OCH3), 38.9 (3-C). Calc. for C_17_H_16_O_4_, [M + H]^+^ *m*/*z*: 285.11214 Da, found 285.11258 Da, [M + Na]^+^ *m*/*z*: 307.09408 found : 307.09413.

General Procedure for flavanones synthesis by base activation (**4B–L**).

To a solution of chalcones **3B–L** (1 eq) in methanol (20 mL) was added 5 eq of sodium acetate. The mixture was heated to reflux for 24 h and was monitored by TLC. The solvent was evaporated, EtOAc (20 mL) was added, and the mixture was washed with distilled water (3 × 20 mL). The solution was dried over anhydrous MgSO_4_ and filtered. The solvent was evaporated under reduced pressure, and the residue was purified by column chromatography (CC) using silica gel 60, 0.2–0.5 mm C35-70mesh ASTH CAS: 7631-86-9 (Scharlau Spain) as stationary phase to give flavanones **4B–L**.

5,7-dimethoxy-2-(3-methoxyphenyl)-2,3-dihydro-4H-benzopyran-4-one or 5,7,3′-trimethoxy-flavanone (**4B**)

Prepared following the general procedure starting from chalcone **3B** (0.603 g, 1.92 mmol). This product was purified by column chromatography (C_6_H_12_/EtOAc 90:10, 70:30, 50:50). Pale yellow solid (0.445 g, 1.41 mmol), 74% yield, m.p. (Electrothermal apparatus): 89–91 °C.

^1^H NMR (500 MHz, CDCl_3_): δ 7.32 (t, *J* = 7.9 Hz, 1H, 5′-H), 7.02 (m, 2H, 2′-H, 6′-H), 6.90 (dd, *J =* 8,3 Hz, 2.1 H*z*, 1H, 4′-H), 6.16 (d, *J* = 2.3 Hz, 1H, 6-H), 6.09 (d, *J* = 2.3 Hz, 1H, 8-H), 5.38 (dd, *J* = 13.3 Hz, 2.8 *Hz*, 1H, 2-H), 3.89 (s, 3H, 7-OCH_3_), 3.83 (s, 3H, 5-OCH_3_), 3.82 (s, 3H, 3′-OCH_3_), 3.00 (dd, *J* = 16.5 Hz, 13.3 Hz, 1H, 3eq-H), 2.79 (dd, *J* = 16.50 Hz, 2.80 Hz, 1H, 3ax-H), ^13^C NMR (125 MHz, CDCl_3_): δ 189.2 (4-C, CO), 166.0 (7-C), 164.9 (5-C), 162.3 (8a-C), 159.9 (3′-C), 129.89 (1′-C), 129.88 (5′-C), 118.3 (6′-C), 114.0 (4′-C), 111.8 (4a-C), 106.1 (2′-C), 93.6 (6-C), 93.2 (8-C), 79.1 (2-C), 56.2 (7-OCH3), 55.6 (5-OCH_3_), 55.3 (3′-OCH_3_), 45.6 (3-C). HRMS (ESI): Calc. for C_18_H_18_O_5_, [M + H]^+^ *m*/*z*: 315.12270 Da, found 315.12292, [M + Na]^+^ *m*/*z*: 337.10464, found: 337.10455.

5,7-dimethoxy-2-(4-methoxyphenyl)-2,3-dihydro-4H-benzopyran-4-one or 5,7,4′-trimethoxy-flavanone (**4C**)

Prepared following the general procedure starting from chalcone **3C** (0.32 g, 1.08 mmol). This product was purified by column chromatography (C_6_H_12_/EtOAc 70:30). Pale yellow solid (0.213 g, 0.68 mmol), 66% yield, m.p. (Electrothermal apparatus): 60–70 °C.

^1^H-NMR (400MHz, CDCl_3_): δ 7.44–7.34 (m, 2H, 2′-H, 6′-H), 7.01–6.91 (m, 2H, 3′-H, 5′-H), 6.16 (d, *J* = 2.3 Hz, 1H, 6-H), 6.11 (d, *J* = 2.3 Hz, 1H, 8-H), 5.38 (dd, *J* = 13.1 Hz*, 2.9 Hz,* 1H, 2-H), 3.92 (s, 3H, 7-OCH_3_), 3.85 (s, 3H, 5-OCH_3_), 3.84 (s, 3H, 4′-OCH_3_), 3.06 (dd, *J* = 16.5 Hz, 13.1 Hz, 1H, 3eq-H), 2.79 (dd, *J* = 16.5 Hz, 2.9 Hz, 1H, 3ax-H). ^13^C NMR (100 MHz, CDCl_3_): δ 189.5 (4-C, CO), 165.9 (7-C), 165.1 (5-C), 162.3 (8a-C), 159.9 (4′-C), 130.8 (1′-C), 127.71 (2′-C), 127.70 (6′-C), 114.17 (3′-C), 114.16 (5′-C), 106.0 (4a-C), 93.6 (6-C), 93.1 (8-C), 79.0 (2-C), 56.2 (7-OCH_3_), 55.6 (5-OCH_3_), 55.4 (4′-OCH_3_), 45.4 (3-C). HRMS (ESI): Calc. for C_18_H_18_O_5_, [M + H]^+^ *m*/*z*: 315.12270 Da, found: 315.12282, [M + Na]^+^ *m*/*z*: 337.10464, found: 337.10436.

2-(4-bromophenyl)-5,7-dimethoxy-)-2,3-dihydro-4H-benzopyran-4-one or 4′-bromo-5,7-dimethoxy-flavanone (**4D**)

Prepared following the general procedure starting from chalcone **3D** (0.1033 g, 0.28 mmol). This product was purified by column chromatography (C_6_H_12_/EtOAc 70:30). Pale yellow solid (0.0719 g, 0.20 mmol), 69% yield, m.p. (Electrothermal apparatus): 60–70 °C.

^1^H NMR (500 MHz, CDCl_3_): δ 7.62–7.54 (m, 2H, 2′-H, 6′-H), 7.38–7.33 (d, *J* = 8.3 Hz, 2H, 3′-H, 5′-H), 6.17 (d, *J* = 2.3 Hz, 1H, 6-H), 6.13 (d, *J* = 2.3 Hz, 1H, 8-H), 5.40 (dd, *J* = 12.8 Hz, 3.2 Hz, 1H, 2-H), 3.92 (s, 3H, 5-OCH_3_), 3.86 (s, 3H, 7-OCH_3_), 2.99 (dd, *J* = 16.5 Hz, 12.8 Hz, 1H, 3ax-H), 2.81 (dd, *J* = 16.5 Hz, 3.2 Hz, 1H, 3eq-H). ^13^C NMR (125 MHz, CDCl_3_): δ 192.3 (4-C, CO), 168.5 (7-C), 166.4 (5-C), 162.5 (8a-C), 134.5 (1′-C), 132.11 (3′-C), 132.10 (5′-C), 129.69 (2′-C), 129.68 (6′-C), 124.2 (4′-C), 106.3 (4a-C), 93.8 (8-C), 91.3 (6-C), 77.3 (2-C), 55.9 (7-OCH_3_), 55.6 (5-OCH_3_). Calc. for C_17_H_15_BrO_4_, [M + H]^+^ *m*/*z*: 363.02265 Da, found 363.02329, [M + Na]^+^ *m*/*z*: 385.00459 found: 385.00514.

2-(4-chlorophenyl)-5,7-dimethoxy-)-2,3-dihydro-4H-benzopyran-4-one or 4′-chloro-5,7-dimethoxy-flavanone (**4E**)

Prepared following the general procedure starting from chalcone **3E** (0.127 g, 0.40 mmol). This product was purified by column chromatography (C_6_H_12_/EtOAc 90:10, 70:30, 50:50). Colorless solid (0.084g, 0.26 mmol), 66% yield, m.p. (Gallenkamp apparatus): 120 °C.

^1^H NMR (400 MHz, DMSO): δ 7.57–7.53 (m, 2H, 2′-H, 6′-H), 7.52–7.47 (m, 2H, 3′-H, 5′-H), 6.25 (d, *J* = 2.3 Hz, 1H, 6-H), 6.22 (d, *J* = 2.3 Hz, 1H, 8-H), 5.57 (dd, *J* = 12.5 Hz, 3.0 Hz, 1H, 2-H), 3.82 (s, 3H, 7-OCH_3_), 3.79 (s, 3H, 5-OCH_3_), 3.02 (dd, *J* = 16.3 Hz, 12.5 Hz, 1H, 3eq-H), 2.67 (dd, *J* = 16.3 Hz, 3.0 Hz, 1H, 3ax-H). ^13^C NMR (100 MHz, CDCl_3_): δ 187.9 (4-C, CO), 165.9 (7-C), 164.5 (5-C), 162.3 (8a-C), 138.5 (1′-C), 133.4 (4′-C), 128.99 (3′-C), 128.98 (5′-C), 128.86 (2′-C), 128.85 (6′-C), 105.9 (4a-C), 94.2 (6-C), 93.5 (8-C), 77.9 (2-C), 56.2 (7-OCH_3_), 56.4 (5-OCH_3_), 45.1 (3-C). HRMS (ESI): Calc. for C_17_H_15_ClO_4_, [M + H]^+^ *m*/*z* : 319.07316 Da, found 319.07330, [M+ Na]^+^ *m*/*z*: 341.05511, found: 341.05529.

2-(5,7-dimethoxy-4-oxo-3,4-dihydro-2H-benzopyran-2-yl)benzoic acid or 2′-carboxy-5,7-dimethoxy-flavanone (**4F**)

Prepared following the general procedure starting from chalcone **3F** (0.220g, 0.67 mmol). This product was purified by column chromatography (C_6_H_12_/EtOAc 70:30). White solid (0.159 g, 0.48 mmol), 72% yield, m.p. (Gallenkamp apparatus): 168 °C.

^1^H NMR (400 MHz, CDCl_3_): δ 13.75 (s, 1H, 2′-OH), 7.93 (d, *J* = 7.7 Hz, 1H, 3′-H), 7.70–7.65 (m, 1H, 4′-H), 7.63–7.53 (m, 2H, 5′-H, 6′-H), 6.18 (dd, *J* = 7.9 Hz, 5.4 Hz, 1H, 2-H), 6.13 (d, *J* = 2.3 Hz, 1H, 6-H), 5.94 (d, *J* = 2.3 Hz, 1H, 8-H), 3.89 (dd, *J* = 18.5 Hz, 5.3 Hz, 1H, 3ax-H), 3.43 (dd, *J*
*=* 18.5 Hz, 7.9 Hz, 1H, 3eq-H). ^13^C NMR (125 MHz, CDCl_3_): δ 200.2 (4-C, CO), 170.5 (2′-COOH), 167.8 (7-C), 166.7 (5-C), 162.7 (8a-C), 150.3 (2′-C), 134.2 (4′-C), 129.2 (3′-C), 126.0 (1′-C), 125.6 (6′-C), 123.1 (5′-C), 105.7 (4a-C), 93.8 (6-C), 91.1 (8-C), 77.48 (2-C), 55.68 (7-OCH_3_), 55.67 (5-OCH_3_), 48.94 (3-C). Calc. for C_18_H_16_O_6_, [M + H]^+^ *m*/*z*: 329.10197 Da, found: 329.10205, [M + Na]^+^ *m*/*z*: 351.08391, found: 351.08389.

2,3-dihydro-2-phenyl-4H-1-benzopyran-4-one or flavanone (**4G**)

Prepared following the general procedure starting from chalcone **3G** (0.5g, 2.23 mmol). This product was purified by column chromatography (C_6_H_12_/EtOAc 90:10, 70:30, 50:50). Colorless solid (0.294 g, 1.31 mmol), 59% yield, m.p. (Gallenkamp apparatus): 70 °C (75–76 °C, [37]).

^1^H NMR (400 MHz, CDCl_3_): δ 7.97 (dd, *J* = 8.1 Hz, 1.7 Hz, 1H, 5-H), 7.58–7.37 (m, 6H, 2′-H, 3′-H, 4′-H, 5′-H, 6′-H, 6-H), 7.11–7.05 (m, 2H, 7-H, 8-H), 5.52 (dd, *J* = 13.3 Hz, 2.9 Hz, 1H, 2-H), 3.13 (dd, *J* = 16.9 Hz, 13.3 Hz, 1H, 3ax-H), 2.93 (dd, *J* = 16.9 Hz, 2.9 Hz, 1H, 3eq-H). ^13^C NMR (125 MHz, CDCl_3_): δ 191.9 (4-C, CO), 161.6 (8a-C), 138.8 (1′-C), 136.2 (7-C), 128.86 (3′-C, 5′-C), 128.8 (4′-C), 127.1 (5-C), 126.16 (2′-C, 6′-C), 121.6 (6-C), 120.9 (4a-C), 118.1 (8-C), 79.6 (2-C), 44.7 (3-C) [37].

2-(4-chlorophenyl)-2,3-dihydro-4H-benzopyran-4-one or 4′-chloro-flavanone (**4H**)

Prepared following the general procedure starting from chalcone **3H** (1.5 g, 5.80 mmol). This product was purified by column chromatography (C_6_H_12_/EtOAc 70:30). Colorless solid (0.556 g, 2.15 mmol), 37% yield, m.p. (Gallenkamp apparatus): 80–90 °C (93-95 °C, [37]).

^1^H NMR (400 MHz, CDCl_3_): δ 7.96 (ddd, *J* = 7.8 Hz, 1.6 Hz, 0.4 Hz, 1H, 5-H), 7.55 (ddd, *J* = 8.7 Hz, 7.2 Hz,1.8 Hz, 1H, 2′-H), 7.48–7.41 (m, 4H, 3′-H, 5′-H, 6′-H, 6-H), 7.12–7.04 (m, 2H, 7-H, 8-H), 5.50 (dd, *J* = 13.1 Hz, 2.9 Hz, 1H, 2-H), 3.07 (dd, *J* = 16.8 Hz, 13.1 Hz, 1H, 3ax-H), 2.91 (dd, *J* = 16.8 Hz, 2.9 Hz, 1H, 3eq-H). ^13^C NMR (100 MHz, CDCl_3_): δ 191.5 (4-C, CO), 161.6 (8a-C), 137.3 (1′-C), 136.3 (4′-C), 133.86 (6-C), 133.85 (7-C), 129.1 (3′-C, 5′-C), 127.52 (2′-C), 127.51 (6′-C), 127.1 (5-C), 121.8 (4a-C), 118.1 (8-C), 78.8 (2-C), 44.6 (3-C) ([37]).

2,3-dihydro-2-(4-methoxyphenyl)-4H-1-benzopyran-4-one or 4′-methoxy-flavanone (**4I**)

Prepared following the general procedure starting from chalcone **3I** (1 g, 3.93 mmol). This product was purified by column chromatography (C_6_H_12_/EtOAc 90:10, 70:30, 50:50). Pale yellow solid (0.374 g, 1.47 mmol), 37% yield, m.p. (Gallenkamp apparatus): 85 °C (87–88 °C, [37]).

^1^H NMR (400 MHz, CDCl_3_): δ 7.95 (dd, *J* = 8.0 Hz, 1.5 Hz, 1H, 5-H), 7.55–7.50 (m, 1H, 2′-H), 7.46–7.41 (m, 2H, 3′-H, 5′-H), 7.01–6.95 (m, 2H, 6′-H, 6-H), 7.10–7.03 (m, 2H, 7-H, 8-H), 5.46 (dd, *J* = 13.3 Hz, 2.9 Hz, 1H, 2-H), 3.86 (s, 3H, 4′-OCH_3_), 3.13 (dd, *J* = 16.8 Hz, 13.3 Hz, 1H, 3ax-H), 2.89 (dd, *J* = 16.7 Hz, 2.9 Hz, 1H, 3eq-H). ^13^C NMR (100 MHz, CDCl_3_): δ 162.6 (4′-C), 137.2 (1′-C), 129.84 (6-C), 129.83 (7-C), 129.65 (3′-C), 129.64 (5′-C), 125.9 (5-C), 125.95 (2′-C), 125.94 (6′-C), 122.7 (4a-C), 118.8 (8-C), 76.6 (2-C), 43.3 (3-C) [37].

2-(2’-Carboxyphenyl)benzopyran-4-one o r 2′-carboxyflavanone (**4J**)

Prepared following the general procedure starting from chalcone **3J** (0.6 g, 2.24 mmol). This product was purified by column chromatography (C_6_H_12_/EtOAc 70:30). Colorless solid (0.486 g, 1.81 mmol), 81% yield, m.p. (Gallenkamp apparatus): 130 °C.

^1^H NMR (500 MHz, CDCl_3_): δ 12.01 (br s, 2′-COOH) 7.94 (d, *J* = 7.8 Hz, 1H, 5-H), 7.69 (ddd, *J* = 7.5 Hz, 7.5 Hz, 0.8 Hz, 1H, 3′-H), 7.65 (dd, *J* = 8.1 Hz, 1.4 Hz, 1H, 6′-H), 7.60–7.55 (m, 2H, 4′-H, 5′-H), 7.51 (dd, *J* = 8.1 Hz, 1.4 Hz, 1H, 6-H), 7.06 (dd, *J* = 8.4 Hz, 0.7 Hz, 1H, 7-H), 6.91 (ddd, *J* = 7.9 Hz, 7.3 Hz, 0.9 Hz, 1H, 8-H), 6.16 (app t, *J* = 6.5 Hz, 1H, 2-H), 3.80 (dd, *J* = 17.6 Hz, 6.2 Hz, 1H, 3ax-H), 3.47 (dd, *J* = 17.6 Hz, 6.8 Hz, 1H, 3eq-H).^13^C NMR (100 MHz, CDCl_3_): δ 201.6 (4-C, CO), 169.9 (2′-COOH), 162.6 (8a-C), 134.4 (1′-C), 129.6 (4′-C), 129.8 (6-C), 129.6 (7-C), 129.81 (3′-C), 129.80 (5′-C), 125.9 (5-C), 122.68 (2′-C), 122.67 (6′-C), 119.3 (4a-C), 118.4 (8-C), 76.6 (2-C), 43.3 (3-C). Calc. for C_16_H_12_O_4_, [M + H]^+^ *m*/*z*: 269.08084 Da, found: 269.08049, [M + Na]^+^ *m*/*z*: 291.06278, found: 291.06275.

2-(5-bromo-2-methoxyphenyl)-2,3-dihydro-4H-benzopyran-4-one or 5′-bromo-2′-methoxy-flavanone (**4K**)

Prepared following the general procedure starting from chalcone **3K** (1.5 g, 4.50 mmol). This product was purified by column chromatography (hexane/dichloromethane 90:10, 70:30, 50:50, 70:30 then ethyl acetate and methanol). White solid (0.846 g, 0.25 mmol), 56% yield, m.p. (Gallenkamp apparatus): 130–135 °C.

^1^H NMR (400 MHz, CDCl_3_): δ 7.97 (dd, *J* = 7.8 Hz, 1.8 Hz, 1H, 5-H), 7.80 (dd, *J* = 2.5 Hz, 0.6 Hz, 1H, 6′-H), 7.54 (ddd, *J* = 8.9 Hz, 7.1 Hz, 1.7 Hz, 1H, 8-H), 7.46 (dd, *J* = 8.9 Hz, 2.5 Hz, 1H, 4′-H), 7.13–7.06 (m, 2H, 6-H, 7-H), 6.82 (d, *J* = 8.8 Hz, 1H, 3′-H), 5.80 (dd, *J* = 13.4 Hz, 2.8 Hz, 1H, 2-H), 3.85 (s, 3H, 2′-OCH_3_), 3.00 (dd, *J* = 16.8 Hz, 13.4 Hz, 1H, 3ax-H), 2.85 (dd, *J* = 16.8 Hz, 2.8 Hz, 1H, 3eq-H). ^13^C NMR (100 MHz, CDCl_3_): δ 192.2 (4-C,CO), 161.7 (8a-C), 154.7 (2′-C), 136.1 (4′-C), 131.9 (6′-C), 129.8 (6-C), 129.3 (7-C), 127.1 (5-C), 121.7 (4a-C), 120.9 (1′-C), 118.1 (5′-C), 112.3 (3′-C, 8-C), 74.2 (2-C), 55.6 (2′-OCH_3_), 43.7 (3-C). Calc. for C_16_H_13_BrO_3_, [M + H]^+^ *m*/*z*: 333.01208 Da, found: 333.01236, [M + Na]^+^ *m*/*z*: 354.99403, found: 354.99396.

5-methoxy-2-phenyl-2,3-dihydro-4H-benzopyran-4-one or 5-methoxy-flavanone (**4L**)

Prepared following the general procedure starting from chalcone **3L** (0.657 g, 2.58 mmol). This product was purified by column chromatography (C_6_H_12_/EtOAc 90:10, 70:30 and 50:50). Colorless solid (0.056 g, 0.2 mmol), 9% yield, m.p. (Electrothermal apparatus): 95–98.7 °C.

^1^H NMR (500 MHz, CDCl_3_): δ 7.49–7.34 (m, 6H, 2′-H,3′-H, 4′-H, 5′-H, 6′-H, 7-H), 6.66 (dd, *J* = 8.4 Hz, 0.6 Hz, 1H, 8-H), 6.55 (d, *J* = 8.3 Hz, 1H, 6-H), 5.44 (dd, *J* = 13.2 Hz, 2.9 Hz, 1H, 2-H), 3.93 (s, 3H, 5-OCH_3_), 3.07 (dd, *J* = 16.4 Hz, 13.2 Hz, 1H, 3ax-H), 2.86 (dd, *J* = 16.4 Hz, 2.9 Hz, 1H, 3eq-H). ^13^C NMR (125 MHz, CDCl_3_): δ 190.7 (4-C, CO), 163.2 (5-C), 160.8 (8a-C), 138.7 (1′-C), 136.1 (7-C), 128.81 (3′-C, 5′-C), 128.68 (4′-C), 126.11 (2′-C, 6′-C), 111.4 (4a-C), 110.2 (8-C), 104.1 (6-C), 78.9 (2-C), 56.2 (5-OCH_3_), 45.9 (3-C). Calc. for C_16_H_14_O_3_, [M + H]^+^ *m*/*z*: 255.10157 Da, found: 255.10199 Da, [M + Na]^+^ *m*/*z*: 277.08352, found: 277.08392.

### 3.2. Biological Assays

#### 3.2.1. Cell Culture and Treatments

The murine macrophage RAW 264.7 cell line (ATCC, TIB-71) was used as an in vitro model for studying the immunomodulatory properties of the molecules. Cells were cultured in Dulbecco’s Modified Eagle’s Medium (DMEM) GlutaMAX™ (Gibco; 10566016) containing 4.5 g/L D-glucose, HEPES buffer at 25 mM, and supplemented with 10% fetal bovine serum (FBS; Gibco), antibiotics, and antifungal solution (penicillin, 10,000 U; streptomycin, 10 mg/mL; amphotericin B, 25 µg/mL). Cell cultures were maintained by cyclic resuspension in 75 cm^2^ culture flasks in a humidified atmosphere with 5% CO_2_ at 37 °C. For biological assays, cells were seeded at 150,000 cells/well in 96-well plates and left to recover for 24 h before treatments. The cells were incubated for an additional 24 h with LPS at 1µg/mL and/or pinocembrin (Sigma–Aldrich, USA) or molecules in dimethyl sulfoxide (DMSO; Sigma Aldrich) at 0.1% and at the final concentration, as indicated in Figure 3, Figure 4 and Figure 5. Control cells were incubated under the same conditions with or without DMSO at 0.1%, without LPS or molecules. Dexamethasone (Sigma–Aldrich; D4902-100MG) at 100 nM was used as the reference anti-inflammatory molecule inhibiting NO production ([38]). Cell culture supernatants were collected for nitrite quantification and cytotoxicity analyses.

#### 3.2.2. Determination of Cell Mortality

Cytotoxicity was evaluated by quantifying the release of lactate dehydrogenase (LDH) in the culture supernatant that correlates with the amount of cell death and membrane damage, providing an accurate measurement of cellular toxicity [39]. LDH was quantified using the commercial CytoTox 96^®^ Non-Radioactive Cytotoxicity Assay (Promega; G1780) following the manufacturer’s specifications. Absorbance at 450 nm (A_450_) was read using a microplate spectrophotometer (Multiskan™ FC, Thermo Fisher Scientific). LDH in the supernatants was normalized against absorbance obtained for total lysed cells, and results were expressed as percent of cytotoxicity, as recommended for LDH-based assays [40].

#### 3.2.3. Quantification of Nitric Oxide (NO)

Nitrite (NO_2_^™^) production was determined as an indicator of Nitric Oxide (NO) synthesis in cell culture supernatant, as previously described [41], using the Griess Reagent System (Promega; G2930) and following the supplier’s recommendations. Briefly, 50 µL of Griess reagent A (1% sulfanilamide in 5% phosphoric acid) was added to 50 µL of cellular supernatant and incubated for 5–10 min at room temperature, protected from light, before additional dispensing of Griess reagent B (0.1% N-1-napthylethylenediamine dihydrochloride in water). Absorbance at 570 nm (A_570_) was read using a microplate spectrophotometer (Multiskan™ FC, Thermo Fisher Scientific). Standard calibration curves were prepared using the provided sodium nitrite and after serial dilutions (0–100 µM). Due to the good correlation factor of this linear regression (R^2^ = 0.98), the nitrite concentrations accumulated in the cellular supernatants was determined. The reduction or increase in nitrite levels was assessed, and results for molecules are expressed as the percentage of inhibitory response of nitrite production in the target sample relative to the LPS-induced nitrite production level, calculated as the inhibition of NO produced according to the following equation:% Inhibitory response = 100 × [1 − ([NO]_M+LPS_ − [NO]_Blank_)/([NO]_LPS_ − [NO]_Blank_)]
where [NO]_Blank_, [NO]_LPS_, and [NO]_M+LPS_ are the concentrations of NO quantified when cells were not treated, induced with LPS, and treated with the molecules and LPS, respectively.

#### 3.2.4. Predictive Analysis of Drug-Like Absorption

Log P predictions were performed using the software ACD/ChemSketch (ACD/Labs). Molecular properties were determined using the Molinspiration Cheminformatics website (http://www.molinspiration.com/cgi-bin/properties, accessed on 26 February 2022).

#### 3.2.5. Statistical Analysis

Results are provided as mean +/− standard deviation (SD). Statistical analyses were performed using the software statistical package Prism 9.0 (GraphPad Software LLC, USA). Cytotoxicity and anti-inflammatory inhibitory response were evaluated using a Mann–Whitney nonparametric test to compare distribution between treatments and LPS induction. *p* values ≤ 0.05 were considered significant. The concentrations of molecules leading to 50% of inhibitory activity (IC_50_) were also calculated using the software Prism 9.0 from a logarithm-transformed (inhibitor) vs. normalized response equation.

## 4. Conclusions

Twelve flavanone derivatives were successfully synthesized and were evaluated for their ability to inhibit NO production against commercial pinocembrin. Six of these derivatives were found to be highly active, with percentage inhibition responses greater than 50%. The most active flavanone derivatives were characterized by the presence of a carboxyl group in the *ortho*-position or a bromo-group in the *para*-position of the B-ring. The importance of the presence of methoxy-groups on the inhibition of NO production was noted, in line with findings by Lee in 2015. In particular, the incorporation of a methoxy group on ring A decreased the biological activity. In contrast, the least active molecules were 4’-chloro flavanone, 4’-methoxyflavanone, and pinocembrin. SAR identified flavanone **4F** carrying a carboxyl group in the *ortho*-position of the B-ring as a potential new lead compound for anti-inflammatory activity.

## Figures and Tables

**Figure 1 molecules-27-01781-f001:**
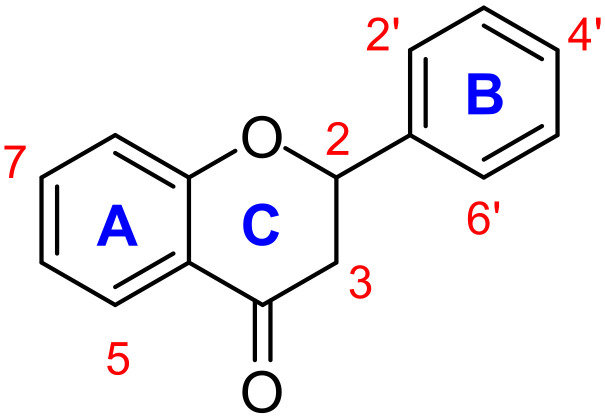
General structure of flavanone.

**Figure 2 molecules-27-01781-f002:**
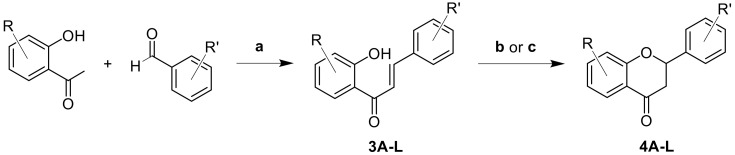
Synthesis of chalcone (**3A–3L**) and flavanone (**4A–4L**) derivatives. Reaction conditions: (**a**) excess of NaOH in 25 mL of MeOH; (**b**) basic activation: 5 eq NaOAc in MeOH (20 mL) under reflux; (**c**) photochemical method: in EtOH at 419 nm, 30 °C for 7 days.

**Figure 3 molecules-27-01781-f003:**
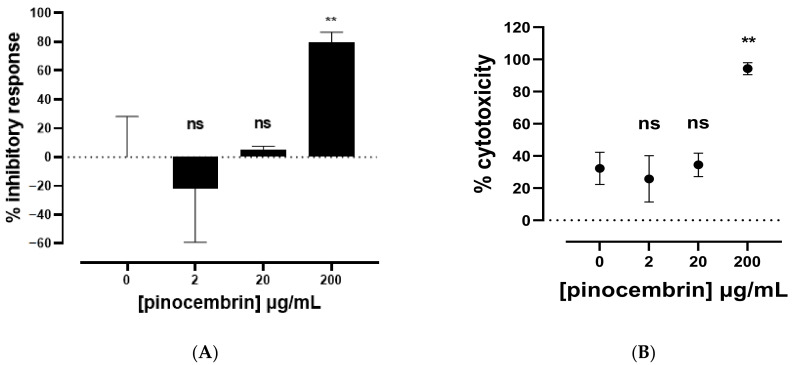
(**A**) Anti-inflammatory activity. Nitrite representative of NO production was quantified using Griess reagent, and percent of the inhibitory response was calculated compared to the level of LPS-dependent nitrite production; (**B**) cytotoxicity of pinocembrin (PC) on LPS-induced RAW264.7. Murine macrophages RAW264.7 (150,000 cells/well in P96) were treated with LPS (*E. coli* 0111:B4) at 1 µg/mL and pinocembrin at 2, 20, and 200 µg/mL for 24 h. Cytotoxicity was measured by quantification of LDH production. Results are means ± SD (at least *n* = 3 well replicates). Mann–Whitney test was used to compare LPS and molecule treatments. ns, nonsignificant; **, *p* < 0.01.

**Figure 4 molecules-27-01781-f004:**
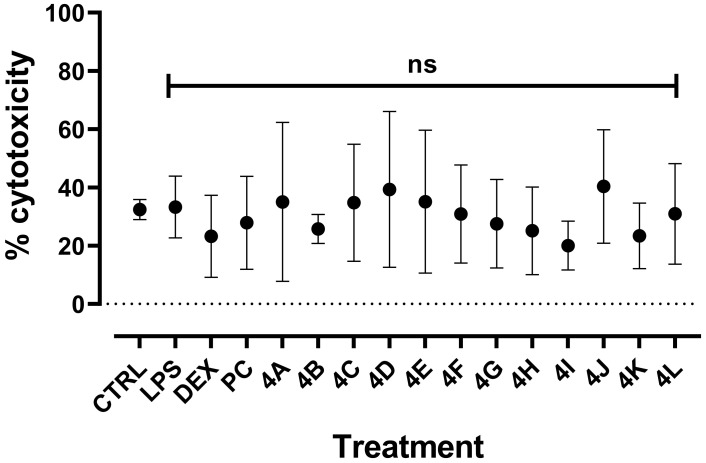
Cytotoxicity of all derivatives on LPS-induced RAW264.7. Murine macrophages RAW264.7 (150,000 cells/well in P96) were treated with LPS (*E. coli* 0111:B4) at 1 µg/mL, analogues and pinocembrin (PC) at 2 µg/mL (in DMSO 0.1%), or anti-inflammatory dexamethasone (DEX) at 100 nM for 24 h. Cytotoxicity was measured by quantification of LDH production. Results are means ± SD of 4 independent experiments (with *n* = 3 well replicates/experiment). Mann–Whitney test was used to compare LPS and molecule treatments. ns, nonsignificant.

**Figure 5 molecules-27-01781-f005:**
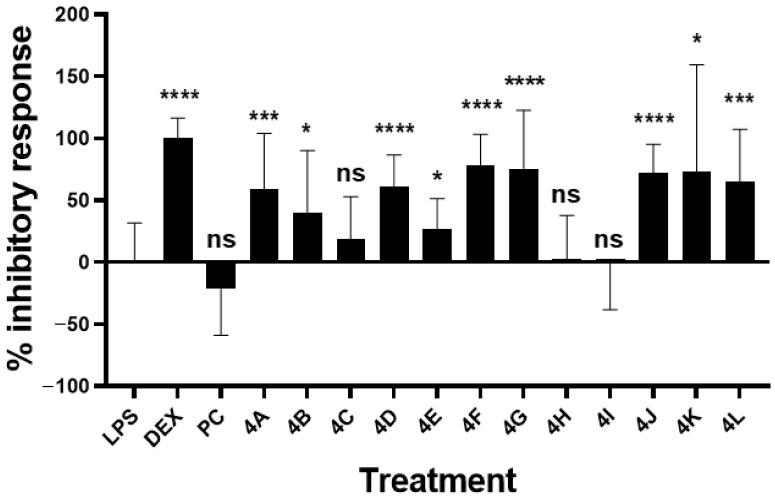
Inhibitory effect of analogues on LPS-induced NO production by RAW264.7. Murine macrophages RAW264.7 (150,000 cells/well in P96) were treated with LPS (*E. coli* 0111:B4) at 1 µg/mL, analogues and pinocembrin (PC) at 2 µg/mL (in DMSO 0.1%), or anti-inflammatory dexamethasone (DEX) at 100 nM for 24 h. Nitrite representative of NO production was quantified using Griess reagent, and percent of the inhibitory response was calculated compared to the level of LPS-dependent nitrite production. Bars represent mean ± SD of 4 independent experiments (with *n* = 3 well replicates/experiment). Mann–Whitney test was used to compare LPS and molecule treatments. ns, nonsignificant; *, *p* < 0.05; ***, *p* < 0.0005; ****, *p* < 0.0001.

**Figure 6 molecules-27-01781-f006:**
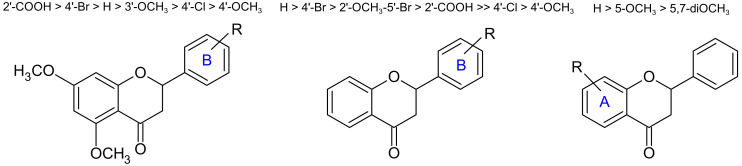
Correlations between structures and anti-inflammatory activity.

**Figure 7 molecules-27-01781-f007:**
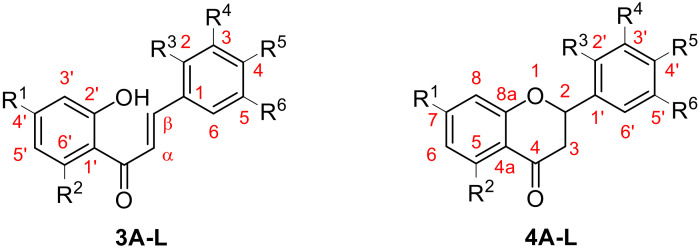
Structures of chalcone and flavanone derivatives.

**Table 1 molecules-27-01781-t001:** Synthesis of chalcones **3A–3L** and flavanone derivatives **4A–4L**.

Compound	R	R’	Chalcones 3	Flavanones 4
			Yield (%)	Yield (%)
**A**	5, 7-(OCH_3_)_2_	H	27	7 *
**B**	5, 7-(OCH_3_)_2_	3′-OCH_3_	82	74
**C**	5, 7-(OCH_3_)_2_	4′-OCH_3_	92	66
**D**	5, 7-(OCH_3_)_2_	4′-Br	52	69
**E**	5, 7-(OCH_3_)_2_	4′-Cl	23	66
**F**	5, 7-(OCH_3_)_2_	2′-COOH	36	72
**G**	H	H	62	59
**H**	H	4′-Cl	86	37
**I**	H	4′-OCH_3_	56	37
**J**	H	2′-COOH	32	81
**K**	H	2′-OCH_3_, 5′-Br	84	56
**L**	5-OCH_3_	H	30	9

Note: * 98% conversion after irradiation with 419 nm light for 7 days.

**Table 2 molecules-27-01781-t002:** Inhibitory effect of analogues **4A–L** on LPS-induced NO production by RAW264.7.

Compound	Inhibitory Response (%) ^a^	*p* Value ^b^
Pinocembrin (PC)	−21.81 ± 37.42	0.1732 ^ns^
Dexamethasone	100.4 ± 16.14	<0.0001 ****
**4A**	58.99 ± 45.19	0.0009 ***
**4B**	40.45 ± 49.71	0.0272 *
**4C**	19.14 ± 33.83	0.1470^ns^
**4D**	61.10 ± 25.50	<0.0001 ****
**4E**	26.85 ± 24.43	0.0318 *
**4F**	78.65 ± 24.73	<0.0001 ****
**4G**	75.65 ± 46.88	<0.0001 ****
**4H**	3.128 ± 34.64	0.5605 ^ns^
**4I**	−0.87 ± 37.54	0.8116 ^ns^
**4J**	72.56 ± 22.70	<0.0001 ****
**4K**	73.29 ± 86.10	0.0105 *
**4L**	64.97 ± 42.37	0.0003 ***

Note: **^a^** Percentage of inhibitory response of compounds used at 2µg/mL compared to the level of LPS-dependent nitrite production in RAW264.7 cells expressed in mean ± SD; **^b^** statistical analyses were conducted. Mann–Whitney test was used to compare LPS and molecule treatments. ^ns^, nonsignificant; *, *p* < 0.05; ***, *p* < 0.0005; ****, *p* < 0.0001.

**Table 3 molecules-27-01781-t003:** Inhibitory response of PC and selected derivatives on NO produced by LPS-induced RAW264.7.

Compound	IC_50_ (µg/mL) ^a^	95% CI ^b^
Pinocembrin (PC) ^c^	203.60 ^c^	101.30−569.31 ^c^
Dexamethasone	0.005	0.003−0.008
**4D**	1.030	0.675−1.382
**4F**	0.906	0.550−1.765
**4G**	0.603	0.366−1.003
**4J**	1.830	1.467−2.677

Note: **^a^** Concentrations of molecules leading to 50% of inhibitory activity calculated using a logarithm-transformed (inhibitor) vs. normalized response equation; **^b^** 95% of Confidence Interval (CI); ^c^ [26].

**Table 4 molecules-27-01781-t004:** Predictive analysis of drug-like absorption.

Compound	IC_50_ (µg/mL)	M_W_ (Da)	Log *p*	Hydrogen Bond Acceptors	Hydrogen Bond Donors	Polar Surface Area (Å^2^)
**4D**	1.030	363.21	4.32 ± 0.41	4	0	44.77
**4F**	0.906	328.32	3.23 ± 0.38	6	1	82.07
**4G**	0.603	224.26	3.62 ± 0.26	2	0	26.30
**4J**	1.830	268.27	3.29 ± 0.27	4	1	63.60

## Data Availability

Not applicable.

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
