# Peer review of "Synthesis and Investigation of Flavanone Derivatives as Potential New Anti-Inflammatory Agents"

_molecules, 2022, doi:10.3390/molecules27061781_

Round 1
Reviewer 1 Report
Authors in this manuscript describe the synthesis and their investigation of flavanone derivatives as potential new anti-inflammatory agents. This manuscript reviews 38 articles and provide a complex survey of current literature dealing with this topic. The topic of this manuscript is up to date, interesting and well suited for journal Molecules. The manuscript is well written and divided into 5 parts, the text is clear and easy to read. For better understanding authors used 7 illustrations and 3 tables. This aid readers understanding. I suggest checking for some small spelling mistakes and grammar errors. Otherwise, I have no major concerns about this manuscript and I recommend it for publication.
Author Response
Response to Reviewer 1 Comments :
Point 1: I suggest checking for some small spelling mistakes and grammar errors
Response 1: Thank you for your time and your work on the manuscript reviewing. Spelling mistakes and grammar errors have been checking. Corrections are indicated in red in the text.
Reviewer 2 Report
Dear authors,
Regarding the manuscript with ID molecules-1601144 titled “Synthesis and Investigation of Flavanone Derivatives as Potential New Anti-Inflammatory Agents”.
The manuscript is of current interest in the fields of phytochemistry and the potential therapeutic applications of natural products.
The authors present a manuscript that deals with the synthesis of flavanone derivatives to evaluate their structure-activity relationship regarding their anti-inflammatory properties. In this sense, flavonoids have been studied as potential anti-inflammatory agents; many studies focus on the evaluation of flavonoids from plant extracts, which shows their bioactive potential but there is still a lack of knowledge on their specific mechanisms of action and how they interact with cellular targets.
To assess this problematic, the authors the authors synthesized 2,3-dihydroflavanone derivatives with different functional groups and evaluated their effect on the production of nitric oxide in RAW 264.7 macrophages, which are commonly used in inflammation-related assays. The manuscript shows sound research, with appropriate controls, analysis of results and their discussion.
in conclusion, the authors mention that the flavanone derivatives with highest anti-inflammatory activity are flavanone, 2’-carboxy-5,7-dimethoxy-flavanone, 4’-bromo-5,7-dimethoxy-flavanone, and 2’-carboxylflavanone. Although it is not the aim of the manuscript, it will provide more information to the potential readers a quick predictive analysis of their drug-like absorption by checking their chemical characteristics with the rule of 5.
Author Response
Response to Reviewer 2 Comments:
Point 1: Although it is not the aim of the manuscript, it will provide more information to the potential readers a quick predictive analysis of their drug-like absorption by checking their chemical characteristics with the rule of 5.
Response 1: Thank you for your time and your work on the manuscript reviewing. Predictive analysis of drug-like absorption have been added in table 4 and lines 242-245 ; 716-719
Reviewer 3 Report
This manuscript represents an interesting study in which the authors prepared twelve 2,3-dihydroflavanone derivatives and evaluated their cytotoxic effect and anti-inflammatory activity on LPS stimulated mouse macrophage cell line. They identified several compounds with strong inhibitory activity on NO concentration as comparing to pinocembrin.
The study is very well performed and methodological approaches are described in details. Results clearly demonstrated the level of anti-inflammatory effect of each compound.
I have question to authors related to cytotoxicity which was quantified by LDH production
Authors found that “LPS related cytotoxicity was determined at 33.26% ± 10.61% and no significant increase was observed for any flavanone at 2 µg/mL, with percentages of cytotoxicity ranging between 20 to 40%“
Do you have explanation why as many as 33, 2% of cells (upon activation to inflammatory phenotype following LPS) after 24 h of incubation were dead or damaged? Have you evaluated cell mortality in unstimulated cells? Was similar rapid decrease of viability reported by other authors?
It seems that, although nonsignificant, some compounds, for example 4I showed moderate cytoprotective effect, what is interesting.
Author Response
Response to Reviewer 3 Comments
Thank you for your time and your work on the manuscript reviewing
Point 1: Authors found that “LPS related cytotoxicity was determined at 33.26% ± 10.61% and no significant increase was observed for any flavanone at 2 µg/mL, with percentages of cytotoxicity ranging between 20 to 40%“
Do you have explanation why as many as 33, 2% of cells (upon activation to inflammatory phenotype following LPS) after 24 h of incubation were dead or damaged? Have you evaluated cell mortality in unstimulated cells? Was similar rapid decrease of viability reported by other authors? It seems that, although nonsignificant, some compounds, for example 4I showed moderate cytoprotective effect, what is interesting.
Response 1: As mentioned by Reviewer#3, potential cytotoxicity of molecules was evaluated on RAW264.7 cells under LPS-induced condition as anti-inflammatory effects were investigated upon LPS treatment. Although quantified in each experiments, cytotoxicity of cells without treatment was not reported in the manuscript as supposed not to be needed. However, we included this data in the revised manuscript to better answer to the Reviewer#3’s comments (see new Figure 4 and lines 153-155 in Results).
Literatures about cytotoxicity introduced various types of data as different methods of calculations could possibly be used. Indeed, several studies reported optical density or absolute quantity for LDH measurement instead of percent calculation while other mentioned % of viability. To allow comparison, we included a new reference in Material and Method that better explained calculation used in our study (line 691).
Cells treated with LPS for 24h showed 33.2% +/- 10.61% of cytotoxicity while cytotoxicity for untreated cells was 32.41% +/- 3.4% with no significant differences with LPS. Similar level of cytotoxicity in untreated RAW264.7 cells was previously quantified in the study of Forest and al. also included in the revised manuscript. This relatively high initial mortality could be explained by the experimental parameters. To allow optimal detection of nitrite that could be difficult to be quantified as the detection threshold is low, cells were seeded at 150,000 cells/well and recovered for 24h before additional 24h incubation time. Initial quantify of cells and time of incubation could explained mortality in control cells.
Reviewer 4 Report
The authors synthesized 12 flavanone derivatives to be used as anti-inflammatory agents. The manuscript was well written and designed, and it has some scientific merits. However, there are a few points that should be clarified.
- Concerning the inhibitory effect of analogues 4A-L on LPS-induced NO, in Table 2, all compounds have big SD values. This means the variation among the replicates was very high and this gives some doubts about the accuracy of the method used. Please give an explanation.
- Lines 180-182: How come 4L with (64.97%) inhibitory activity followed 4D (61.10%). Please clarify.
- Line 194: flavanones.
- Lines 297: 305, 309, and 316, Thieury et al., 2017.
- Line 639: at 0.1% final.
- Line 833: 2018.
Author Response
Response to Reviewer 1 Comments
Thank you for your time and your work on the manuscript reviewing
Point 1: Concerning the inhibitory effect of analogues 4A-L on LPS-induced NO, in Table 2, all compounds have big SD values. This means the variation among the replicates was very high and this gives some doubts about the accuracy of the method used. Please give an explanation.
Response 1: We agree with Reviewer#4 regarding the high SD values calculated for the inhibitory effect of the molecules in Table 2 and the Figure 5. Indeed, experiments were reproduced 4 times with 3 replicates for each conditions that explains the variability observed in the results. However, we estimated that the calculation for the statistical analyses were more robust by including all the variability observed. Indeed, we used the Mann-Whitney nonparametric test to compare distribution between treatments and LPS induction. This test specifically compares mean values including SD, and de facto includes SD variation. Moreover, results obtained for the reference anti-inflammatory molecule, dexamethasone (100% of inhibition with the highest P value) and the platform molecule, the pinocembrin at 2µg/mL (no activity at this concentration) were consistent with the literature supporting the validity of our approach, data and calculations.
Point 2: Lines 180-182: How come 4L with (64.97%) inhibitory activity followed 4D (61.10%). Please clarify.
Response 2: We are thankful to Reviewer#4 for this comment as 4D exerts a higher inhibitory activity compared to 4L. Molecule ranking was unfortunately inverted between 4D and 4L in the first version of manuscript. Corrections were made accordingly in the revised version (line 185-186).
Point 3, 4, 5, 6:
- Line 194: flavanones.
- Lines 297: 305, 309, and 316, Thieury et al., 2017.
- Line 639: at 0.1% final.
- Line 833: 2018.
Response 3, 4, 5, 6: We agree, the changes have been made lines 198, 311, 319, 323, 330, 679 and 882